# Efficient Delivery and Replication of Oncolytic Virus for Successful Treatment of Head and Neck Cancer

**DOI:** 10.3390/ijms21197073

**Published:** 2020-09-25

**Authors:** Masakazu Hamada, Yoshiaki Yura

**Affiliations:** Department of Oral and Maxillofacial Surgery, Osaka University Graduate School of Dentistry, Suita, Osaka 565-0871, Japan; narayura630@gmail.com

**Keywords:** oncolytic virotherapy, head and neck cancer, immunogenic cell death, virus delivery, virus replication, tumor antigen, tumor microenvironment

## Abstract

Head and neck cancer has been treated by a combination of surgery, radiation, and chemotherapy. In recent years, the development of immune checkpoint inhibitors (ICIs) has made immunotherapy a new treatment method. Oncolytic virus (OV) therapy selectively infects tumor cells with a low-pathogenic virus, lyses tumor cells by the cytopathic effects of the virus, and induces anti-tumor immunity to destroy tumors by the action of immune cells. In OV therapy for head and neck squamous cell carcinoma (HNSCC), viruses, such as herpes simplex virus type 1 (HSV-1), vaccinia virus, adenovirus, reovirus, measles virus, and vesicular stomatitis virus (VSV), are mainly used. As the combined use of mutant HSV-1 and ICI was successful for the treatment of melanoma, studies are underway to combine OV therapy with radiation, chemotherapy, and other types of immunotherapy. In such therapy, it is important for the virus to selectively replicate in tumor cells, and to express the viral gene and the introduced foreign gene in the tumor cells. In OV therapy for HNSCC, it may be useful to combine systemic and local treatments that improve the delivery and replication of the inoculated oncolytic virus in the tumor cells.

## 1. Introduction

Head and neck cancer refers to cancers developing in the oral cavity, pharynx, and larynx. It is the sixth most common disease, with an annual morbidity of 550,000 and 300,000 deaths worldwide [1,2,3,4,5]. The majority of head and neck cancer is squamous cell carcinoma (SCC). Its etiology is related to smoking and drinking, but in recent years, many oropharyngeal cancers are caused by human papillomavirus (HPV) infection, and when HPV is positive, the prognosis of head and neck SCC (HNSCC) is better than when it is negative [6,7]. The combination of surgery, radiation, and chemotherapy is the standard treatment for HNSCC. However, although there have been advances in these therapies, 5-year survival rates remain 40–50% [1].

In recent years, immunotherapy for cancer has attracted attention as an advanced treatment [8]. In immunotherapy, multifunctional cytokines, such as interleukin (IL)-2, interferon (IFN)-α, and tumor necrosis factor (TNF)-α, were initially used [9,10,11]. Manipulation of the immune system by blocking ligands and receptors that act as regulators of T cell activation, so-called immune checkpoints (ICs), exemplified by the cytotoxic T lymphocyte associated antigen 4 (CTLA-4) and programmed cell death protein 1 (PD-1) and its ligand (PD-L1), has been become an important strategy to control advanced tumors. Immunotherapy became a substantially evaluable therapy as the ICIs received Food and Drug Administration (FDA) approval in 2011 [12,13]. It was also confirmed that antibody therapy against CTLA4 or PD-1 as an immune checkpoint inhibitor (ICI) can maintain its effects for a longer period of time and has fewer side effects than conventional treatments [14]. However, in the KEYNOTE 012 trial for advanced cancer using an ICI, the overall response rate of the patients treated with the anti-PD-1 antibody was 18%, demonstrating that there are cases in which treatment alone is not curative [15]. New immunological therapies, such as antibodies that inhibit other immunosuppressive ligands, vaccine therapy and chimeric antigen receptor (CAR)-T cells, are also being investigated [4,8,16,17,18].

Oncolytic virus (OV) therapy is a treatment that aims to infect the tumor with the virus and destroy the tumor by its cytopathic effects. In addition to the cytopathic effects, OV activates systemic tumor immunity and is expected to be effective for metastatic tumors to which the virus is not directly administered [19,20,21,22,23]. In 2015, a mutant HSV-1, talimogene laherparepvec (T-vec), was approved by the United States FDA for difficult-to-resect melanoma [24,25]. Thereafter, T-vec treatment was extended to other malignancies such as HNSCC, breast cancer, pancreatic cancer and sarcoma [21,26]. OV therapy frequently causes adverse events such as fever, fatigue, nausea, malaise, and increased liver function, which are associated with viral infections. Severe adverse events, including hypotension, tachycardia, cellulitis, dyspnea, and pleural effusion, have also been reported [27]. However, the lack of severe late-onset and permanent dysfunction observed as a sequela of surgery and chemoradiotherapy (CRT) suggests that OV therapy can be used in combination with radiotherapy and chemotherapy. Indeed, the therapeutic effects of T-vec and an ICI on melanoma were doubled compared with the administration of these alone [28,29], suggesting that ICIs can be concomitant agents with OV. Although OV can be administered by intratumoral or intravenous injection, HNSCC develops at a site near the surface, which has the advantage of being treated by direct administration of the virus to the tumor. This review describes the current status of clinical studies of HNSCC using OV, and discusses virus-induced changes in the tumor microenvironment (TME), immunogenic cell death (ICD), tumor-associated antigens, anti-tumor and anti-virus immunity, and improved delivery and replication of administered viruses.

## 2. Current OV Therapy for HNSCC

Viruses currently employed in clinical studies for HNSCC include DNA viruses, such as herpes simplex virus type 1 (HSV-1), vaccinia virus and adenovirus, and RNA viruses such as reovirus, measles virus and vesicular stomatitis virus (VSV) [19,21,26] (Table 1). In these viruses, virulence genes were deleted to reduce their pathogenicity, the virus was altered to selectively infect tumors and a foreign gene, such as a cytokine gene, was added to increase tumor immunity. As RNA viruses, virus strains that are originally low in pathogenicity, tumor-selective, and highly oncolytic are used. The mechanisms of tumor-selectivity of OVs are different in each OV, but RAS expression in tumor cells and the IFN system may play important roles in HSV-1 and reovirus. Other mechanisms are also involved, such as selective replication of thymidine kinase (TK) gene-deleted vaccinia virus in cancer cells and p53 mutation and/or loss of p14arf dependence of an E1B-55kDa gene-deleted adenovirus [23]. The choice of virus for HNSCC depends on whether the administration is systemic or local. In particular, HSV-1 cannot be administered intravenously because it is rapidly neutralized by the existing antibody against HSV-1. 

*Herpesviridae:* HSV-1 is a typical DNA virus that has been investigated as an oncolytic virus and replicates in the nucleus. The majority of adults are infected and usually have antibodies to neutralize free viruses. The genome size is approximately 150 kb, which is sufficient to insert foreign genes [38]. T-Vec is a recombinant human HSV-1 lacking γ34.5 and viral ICP47 genes, which accelerates the expression of US11 gene, and encodes granulocyte-macrophage colony stimulating factor (GM-CSF) [39]. In the treatment of advanced melanoma, the combination of T-Vec with the ICI ipilimumab or pembrolizumab resulted in an objective response rate of 18% for ipilimumab alone versus 39% for the combination. In Phase Ib clinical studies, when patients with advanced melanoma were treated with T-Vec and pembrolizumab, the objective response (OR) and complete response (CR) were 62% and 33%, respectively, which confirmed that T-Vec was more effective when used in combination with an ICI, being a typical example of successful combinational immunotherapy [28,29]. For HNSCC, T-vec injection into metastatic lymph nodes before surgery and CRT promoted the highly degenerative changes on metastatic lymph nodes [30]. Recently, a multicenter phase 1b/3 trial for 36 patients with recurrent or metastatic HNSCC refractory to platinum-based chemotherapy was performed to determine the effects of a combination of T-Vec and pembrolizumab. One dose limiting toxicity (DLT) of T-vec-related fatal arterial hemorrhage was reported. Other than this DLT, there were no treatment-related fatal adverse events. A confirmed partial response (PR) was observed in 5 (13.9%) patients. Ten (27.8%) patients were unevaluable due to early death. The median progression-free survival (PFS) and overall survival (OS) were 3 months (95%CI, 2.0–5.8) and 5.8 months (95% CI, 2.9–11.4), respectively. The efficacy of the combination was similar to that of pembrolizumab monotherapy in historical HNSCC studies [31]. HF10 has a deletion in the UL56 gene and has cell fusion ability. It was previously administered for solid tumors, including pancreatic cancer and HNSCC, and was slightly effective [32,40]. In nine pancreatic cancer patients who completed the treatment, tumor responses were three PR, four stable disease (SD) and two progressive disease (PD) [40]. In preclinical studies, G47Δ-mIL12, a G47ΔHSV-1 expressing murine IL-12, was combined with antibodies against CIs (CTLA-4, PD-1 and PD-L1), overcoming the highly immunosuppressive TME of murine glioblastoma and eradicating the mouse tumors [41]. Inhibition of the TGF-β signaling pathway was reported to be beneficial for OV treatment with HSV-1 [42]. The combined use of EGFR-CAR-transduced NK-92 cells with oncolytic HSV-1 had superior antitumor effects on brain metastases of breast cancer compared with the administration of each alone [43].

*Poxviridae:* As no antibody against vaccinia virus is present in most adults, it can be administered intravenously to infect both primary and distant metastatic lesions. In a phase I clinical study, the safety of GL-ONCI, formally named GLV-1h68, was examined in 19 patients who had locally advanced HNSCC without distant metastases by intravenously viral administration. GL-ONCI was constructed by inserting three expression cassettes encoding Renilla luciferase-Aequorea green fluorescent protein fusion, galactosidase, and β-glucuronidase into F14.5, J2R encoding thymidine kinase and A56R encoding hemagglutinin loci of the genome, respectively [44]. A systemic rash developed as an adverse event. At 30 months of follow-up, the 2-year PFS and OS were 64.1% and 69.2%, respectively [33]. In a phase Ib study, patients with platinum-resistant/refractory recurrent ovarian cancer received repeated intraperitoneal injection of GL-ONCI. Disease control (PR+CR > 15 months) was observed in 55% of the patients [45]. As a preclinical study, vvDD with a double-deletion of the genes encoding TK and the vaccinia growth factor (VGF) was constructed, and its antitumor effects were examined in murine colon and ovarian cancer models in vitro and in vivo. vvDD attracted effector T cells, and induced PD-L1 expression in both cancer and immune cells [46]. Furthermore, vvDD-DAI, which overexpresses the intracellular pattern recognition receptor, the DNA-dependent activator of IFN-regulatory factors (DAI), was prepared in order to increase IFN production and innate and passive immunity. Compared with vvDD alone, administration of vvDD-DAI increased tumor-infiltrating CD8^+^ T cells into the tumor, translating into better efficacy against melanoma in mice [47]. Myxomavirus (MYXV) that belongs to *Poxviridae* family was also altered by deleting the M0111L gene encoding the Bcl-2 homolog from wild-type MYXV, and the lack of Bcl-2 homolog sensitized brain tumor cells to virus-induced cell death and the survival of tumor-bearing mice was prolonged in an immunocompetent model of glioblastoma [48]. Epstein–Barr virus (EBV) is associated with several malignancies, including nasopharyngeal carcinoma. A therapeutic vaccine, MVA-EL, was produced using the modified vaccinia Ankara vector to encode a functional inactive fusion protein of full-length LMP2 and the C-terminal half of EBNA1 [34]. In phase I trials, patients with nasopharyngeal carcinoma received intradermal MVA-EL vaccination. After vaccination, immunity increased to at least one antigen in 8/14 patients (7/14, EBNA1; 6/14, LMP2), including recognition of epitopes [49].

*Adenoviridae*: Although adenovirus is a small DNA virus with a genome size of 26–45 kb, foreign genes can be integrated and a variety of mutant viruses have been constructed [50]. An E1B-55kDa gene-deleted adenovirus, ONYX-015, was developed for treatment of tumors lacking p53 function [35]. In a phase II clinical trials, 37 patients with recurrent HNSCC received intratumoral and peritumoral ONYX-015 injection, and highly selective tissue destruction was observed; significant tumor regression (>50%) was noted in 21% of evaluable patients [35]. A E1 substituted replication-incompetent recombinant adenovirus encoding the p53 gene Ad-p53 was injected intratumorally or intraoperatively in combination with surgery and CRT to treat patients with stage III-IV hypopharyngeal cancer and lymph node metastasis. The OS and disease-free survival (DFS) were significantly extended in the surgery+CRT+rAd-p53 group compared with those in the surgery+CRT group [51]. AdGV.EGR.TNF.11D is a non-replicating adenovirus that expresses human TNF-α under the control of the radiation-inducible promoter (EGR-1). AdGV.EGR.TNF.11D was administered intratumorally and combined with 5-FU and hydroxyurea for the treatment of recurrent HNSCC patients receiving re-irradiation. The response rate was 83.3% and the average survival time was 9.6 months [36]. As a preclinical study, LOAd703 armed with costimulatory CD40L and 4-1BBL was constructed. Infection of LOAd703 was associated with dendritic cell (DC) maturation/expansion, which in turn increased the activation/expansion of NK and tumor-specific T cells, leading to oncolytic activity against pancreatic cancer cells [52]. Delta-24-RGD adenovirus, also called DNX-2401, is a virus in which an RGD peptide motif was introduced into adenovirus fibers to promote interaction with tumor integrins and 24 base pairs were deleted in the E1A gene to restrict replication in tumor cells with abnormalities in the p16/RB/E2F pathway. This was injected to patients with recurrent malignant glioma and resulted in good responses with long-term survival [53]. Bispecific T cell-engagers (BiTEs) represent a new-class of immunotherapeutic molecules consisting of two-single-chain variable fragments (scFv) connected by a flexible linker. One scFv binds to the T lymphocyte marker CD3, whereas the second scFv is directed against one tumor antigen expressed on the surface of tumor cells. By co-engaging T cell effectors and cancer cells, BiTEs can mediate immune-mediated tumor cell lysis [54,55,56,57]. An oncolytic adenovirus, ICOVIR-15K expressing a BiTE antibody targeting EGFR was previously constructed [55]. The virus secretes BiTEs, which bind specifically to CD3^+^ and EGFR+ cells. After the transfer of mononuclear cells to tumor-bearing mice with EGFR-expressing tumor, administration of ICOVIR-15K into the tumor resulted in persistent accumulation of cytotoxic T cell (CTL) and increased antitumor effects. In CAR-T cell therapy, HER2-specific CAR-T cells alone cannot cure solid tumors in the immunosuppressive microenvironment of tumors [18]. The presence of adenovirus-encoded IL-12p70 can prevent the disappearance of HER2- and CAR-expressing T cells at the tumor site. Therefore, adenovirus expressing IL-12p70 and encoding the PD-L1 blocking antibody, CAd12-PDL1, was constructed [18]. When HNSCC xenografts in mice were treated with HER2 and CAR2.CAR-T cells in combination with CAd12-PDL1, the survival rate of the mice was approximately 25 days with either approach, while it was extended to over 100 days by the combined therapy with the control of distant metastasis. 

*Reoviridae*: Reovirus is a double-stranded RNA virus that can grow in RAS-expressing cells. It can be delivered to tumors by intravenous administration and kill tumor cells with few side effects. In a phase I/II study, a total of advanced 31 patients, including 14 with HNSCC, were treated with reovirus type 3 Dearing (RT3D) in combination with taxanes. One (3.8%) patient had CR and 6 (23.1%) had PR, suggesting activity in cancer of the head and neck [37]. A phase II study of reovirus in combination with paclitaxel or carboplatin was conducted for pancreatic adenocarcinoma and melanoma. In the case of pancreatic cancer, the addition of reovirus was not superior to carboplatin/paclitaxel [58]. As a preclinical study in mice, the combination of reovirus with anti-PD-1 antibody enhanced the therapeutic efficacy against subcutaneous melanoma in mice. This showed that checkpoint inhibition increased both NK cell-mediated tumor cell killing and CD8+ T cell antitumor immune responses while reducing Treg activity [59]. 

*Paramyxoviridae:* Measles virus is a minus single-stranded RNA virus. Due to its systemic dissemination properties, intravenous administration can be used as a delivery route. MV-NIS is a detoxified measles virus Edmonton strain that expresses the sodium/iodide symporter (NIS) [60]. CD46 acts as a cell-side receptor during the entry of measles virus and the oncolytic efficacy of MV-NIS is highly correlated with the density of CD46 receptors on target cells. Numerous ions in addition to iodide are transported efficiently by the NIS protein, which enables NIS expression imaging with readily available radioisotopes, such as iodine-123, iodine-125, or technetium-99m, which can be detected by radiation (PET) or single-photon emission computed tomography combined with computed tomography (SPECT/CT) [61]. NIS expression on the surface of infected cells facilitates uptake of radioisotopes, and when used in combination with an isotope, the antitumor effects of MV-NIS are increased in HNSCC cells [62]. 

*Rhabdoviridae*: VSV is a negative-strand RNA virus with low preexisting immunity in humans. VSV has high oncolytic potency and causes strong induction of the innate immune response. VSV encoding the interferon beta transgene (VSV-IFNβ) has increased specificity for tumor cells versus normal healthy tissue. The use of the IFN-β transgene provides tumor specificity by protecting cells whose IFN response is normal and only enables replication in cells with a defective IFN pathway. [63]. In a preclinical study, intratumoral or intravenous administration of VSV-IFNβ resulted in growth delay of SCC and improved survival compared with controls [64], and the anti-tumor function for VSV-IFNβ significantly increased when combined with ICIs in CT26 colon cancer and B16-F10 melanoma mouse models [65].

## 3. Safety and Limitation of OV Therapy

In a recent study with T-vec and pembrolizumab for recurrent or metastatic HNSCC, 20 (55.6%) and 21 (58.3%) patients experienced adverse events related to T-vec and pembrolizumab, respectively. One (6.3%) dose-limiting toxicity (DLT), fatal arterial hemorrhage, was observed among DLT-evaluable patients [31]. When T-Vec is injected to recurrent a tumor that invades an artery, rapid OV-induced tumor destruction can cause arterial hemorrhage. The indications of this therapy for advanced HNSCC should be carefully evaluated. The efficacy of treatment with T-vec and pembrolizumab was not superior to that of pembrolizumab alone, which may indicate the limitation of this combination therapy. In a phase I/II trial involving the infusion of a reovirus for the treatment of advanced malignancies, the grade 3/4 hematological toxicities observed were neutropenia (16.1%), asymptomatic lymphopenia (6.5%), and anemia (3.2%). Nonhematological grade 3/4 toxicities observed included fever (9.7%), myalgia (6.5%), diarrhea (3.2%), nausea (3.2%), vomiting (3.2%), and hypotension (3.2%). Although reovirus alone induced objective clinically meaningful responses in HNSCC [23], adverse events can arise with reovirus combination therapy, showing increased toxicity and decreased quality of life in metastatic colorectal cancer patients. Regarding efficacy, reovirus did not prove superiority in comparison to chemotherapy alone when combined with carboplatin/paclitaxcel in metastatic pancreatic cancer and recurrent ovarian cancer [26]. This suggests the limitation of the combination of reovirus and chemotherapy in the treatment of HNSCC. 

## 4. The TME and OV Infection

In solid tumors, the TME consists of many non-neoplastic cells that are retained or migrate, secretory factors and extracellular matrix proteins. Non-neoplastic cells include cancer-associated fibroblasts (CAFs), adipocytes, stromal cells, vascular endothelial cells, pericytes, lymphatic endothelial cells and immune system cells [66]. Tumor-infiltrating immune cells include NK cells, monocytes, CD8^+^ cytotoxic (memory) T cells, CD4^+^ helper (Th1, Th2) T cells, regulatory T cell (Treg), B-cells, tumor-associated macrophages (TAMs), tumor-associated neutrophils (TANs), myeloid-derived suppressor cells (MDSCs), DCs, and NK cells [66,67,68]. Non-neoplastic cells account for more than 50% of the total tumor mass. The TME also contains elastic collagen, fibronectin fibrils with glycoproteins and the polysaccharides that surround the tumor. The extracellular matrix is not only a scaffold for TME, but also a biological signal from growth factors and chemokines, which are involved in proliferation, migration, and metastasis [69]. The immunosuppressive molecules, including growth factors (e.g., TGF-β), cytokines (IL-10), chemokines, and inflammatory, matrix-remodeling enzymes and metabolites, inhibit the cytotoxic activity of NK and CTL in the TME. Tumors have increased expression of immune evasion targets, including CD47, TGF-β, VEGF, IL-10, FLIP, FAS, and Bcl-xL [21]. In addition, a second immunosuppressive mechanism, an immune checkpoint molecule expressed on the surface of some TME cells, regulates T cell reactivity. The well-known IC-receptors are CTLA4, PD-1, the lymphocyte activation gene-3 (LAG-3), T cell immunoglobulin, mucin domain-containing protein 3 (TIM-3), and T-cell immunoreceptor with Ig and ITIM domains (TIGIT) [70,71]. The corresponding ligands are CD80 and CD86, PD-L1 for CTLA4, and PD-L1 and PD-L2 for PD-1 [72,73,74]. Binding of these IC-receptor-ligands inhibits the anti-tumor immune reaction by cytotoxic T cells and NK cells present in the TME.

When a tumor is infected with OV, antitumor immunity is induced by a different mechanism. OV-mediated cell death is often immunogenic and cellular tumor-specific antigens (TSAs)/tumor-associated antigens (TAAs), danger-associated molecular patterns (DAMPs), viral pathogen-associated molecular patterns (PAMPs) and progeny virus are released, leading to the induction of an inflammatory immune response in the TME. DAMPs include ATP, high mobility group box-1 (HMGB1) and calreticulin [75,76]. In particular, extracellular ATP acts as a “find-me” signal and recruits DCs, and HMGB1 acts as a danger signal ligand for Toll-like receptor 4 and directly activates DCs [77,78]. The calreticulin exposed on the cell surface neutralizes CD47 on the tumor cell surface and promotes phagocytosis as an “eat-me” signal [79]. DAMPs attract antigen presenting cells (APCs), especially DCs to TME, and cause DC to produce inflammatory cytokines, which present TSAs/TAAs, and prime cytotoxic T cells. OV infected cells release type I IFNs and DAMPs simultaneously, eliciting the same immunological effects and expression of certain DAMPs is also activated by type 1 IFN [22]. This suggests that type I IFNs act as DAMPs. Other molecules, such as annexin A1 (ANXA1) and cancer cell-derived nucleic acid, also function as DAMPs [80]. 

In an ideal scenario, TSA/TAA is captured and processed by APC, especially DC. The antigen-bearing DCs migrate to regional lymph nodes, mature and acquire the ability to reprogram T cells. It also induces cancer-specific T cell responses against a broad spectrum of tumor antigens. PAMPs consisting of viral RNA, DNA and proteins are recognized by pattern recognition receptors (PRRs) expressed and sensed by DCs. PRRs include Toll-like receptors, RIG-like receptors, NOD-like receptors and cGAS [81,82,83]. Due to the involvement of PRR, DC produces pro-inflammatory cytokines (e.g., TNF-α and IL-12) and antiviral type I IFN. These cytokines lead to TSA/TAA cross-presentation and priming of CTL. Type I IFNs mature DCs, and up-regulate co-stimulatory CD40 and CD86 molecules that are essential for upregulating the surface expression of MHC class I expression and CTL activity [84,85,86]. 

In melanoma patients who received T-vec intratumorally, Treg and MDSC decreased, and the infiltration of CD8^+^ T cells increased [82,87]. In addition to this, the expression of cytokines, chemokines, and chemokine receptors also increased. Therefore, the release of tumor antigens and the presentation of antigens can function in the activation and proliferation of cytotoxic T cells [82,87,88]. As a model of human oral SCC, when mouse SCCVII and human oral SCC cells were infected with an oncolytic HSV-1 RH2 lacking γ34.5 gene and having cell fusion capacity, the infected SCC cells released ATP, and HMGB1 and CRT were exposed on the cell surface in cell culture, suggesting characteristics of ICD [89]. When RH2 was administered into the tumors in synergistic mice, the antitumor effects of the virus were superior to those in the non-administered tumors formed in the same body, demonstrating that although RH2 induces systemic antitumor immunity, the RH2-induced direct inflammatory reaction plays a role in reducing the size of the tumor [90]. Injection of the culture supernatant of RH2-infected cells exposed to ultraviolet rays to inactivate viruses into cancer-bearing synergistic mice [89] induced tumor-suppressive effects. This suggests that TSA/TAA, DAMPs and PAMPs released from RH2-infected SCC cells in the culture supernatant are responsible for the activation of cytotoxic CD8^+^ T cells through DC maturation [91].

## 5. Tumor Antigens Released by OV Therapy into the TME

Tumor antigens include non-self (virus) and self-antigens [4]. The viral antigens of HPV and EBV themselves can be good targets for induced antitumor immunity against foreign antigenic materials. As HPV E6 and E7 play a central role in carcinogenesis in HNSCC, these HPV antigens can be used in HPV-positive cancers [15].

Non-viral cancer antigens are largely divided into TSAs and TAAs [16,92,93]. TSA is a self-mutated protein (neoepitode) that is expressed only in cancer cells. The neoantigen is not expressed in non-tumor tissues, and is more immunoreactive and less immune-tolerant in non-tumor tissues; therefore, it can be a powerful target for immunotherapy. P53 and RAS are frequently mutated in HPV-negative tumors [17]. Zolkind et al [16] proposed tumor-specific mutant antigen (TSMA). CASP8 is one example of a TSMA identified in human tumors whose mutation may be a driver mutation in HNSCC [94,95]. In these tumors, the mutant protein becomes a target of immune reaction as a neoantigen. TAAs are unmodified self-proteins, which are non-mutated glycosylated proteins MUC1 and CEA or cancer testis antigens (CTAs) [96,97,98]. HER2, MAGE family, BAGE family, SSX family, PRAME and NY-ESO-1 are included in CTAs [99,100,101]. TAA is well expressed in growing tumor cells. MUC1 and CEA have been confirmed in patients, and are being applied clinically. 

Administration of the supernatant of SCCVII cells infected with oncolytic HSV-1 RH2 increased systemic antitumor immunity in mice, suggesting the presence of TSAs/TAAs is the supernatants [89,91]. As a control, proteomics analysis of the culture supernatant of control SCCVII cells revealed the presence of proteins, including fibronectin, thrombospondin-1, heparin sulfate proteoglycan core protein, chondroitin sulfate proteoglycan 4, and type IV collagenase, probably in the form of exosomes. When SCCVII cells were infected with RH2, secretion of these proteins was markedly reduced, whereas the secretion of proteins, including filamin, tubulin, TCP-1 and HSPs, increased. This suggests that DAMPs and chaperon proteins function in the OV-mediated infiltration and activation of antitumor immune cells [102]. As there was no increase in known HNSCC-associated TSAs/TAAs, such as survivin, p53, RAS, HER2, Mart-1, MARG, MUC1, and CEA [17], mutated neoantigens produced during tumor formation may function as TSAs in this SCC model. Even in this situation, OV therapy is an effective method to liberate a wider range of neoantigens in the TME and stimulate tumor-specific immunity.

## 6. Antitumor Immunity and Antiviral Immunity

The purpose of OV therapy is not only to kill cancer cells, but also to overcome the immunosuppressive TME. Infection of tumor cells with OVs creates an inflammatory state by secretion of immune-activating cytokines, changing “cold tumors” to inflammatory “warm tumors” infiltrated with inflammatory cells [103]. IFN production is a major antiviral reaction of cells, which is often dysfunctional in cancer cells. As a result, OVs easily infect transformed cells and exert their cytopathic effects and induce cell degeneration. OV also leads to an antiviral immune response by innate and passive immunity, whether administered intravenously or locally at the tumor [23]. This will result in the rapid elimination of OV and the reduction of OV anti-tumor immunity. Therefore, OV replication and rapid spread within the tumor is necessary for maximal antitumor efficacy before the virus is cleared [104]. In OV therapy, there are many strategies that suppress antiviral immunity and induce a danger signal in view of the effects of antiviral immunity. These include the use of immunomodulators, depletion of antibodies and magnetic nanoparticles [105,106,107,108]. On the other hand, the anti-viral immune responses to OV are beneficial for antitumor immunity because they can overturn the immunosuppressive environment associated with tumors and recruit immune cells to the tumor, leading to virus-induced ICD, thereby activating antitumor immunity. Indeed, with intratumoral OV therapy with Newcastle disease virus (NDV), although pre-existing immunity to NDV limits its replication in tumors, the tumor clearance, abscopal anti-tumor immune cells, and survival were not compromised and were superior in NDV-immunized mice [103,109].

According to previous studies, a special virus can be used to avoid the effects of anti-virus immunity [110]. Human infection with the arenavirus strains lymphocytic choriomeningitis virus (LCMV, strain WE) and Junin virus vaccine (Candid#), which is a clinically applied vaccine virus to protect against Argentine hemorrhagic fever, is usually asymptomatic or causes non-specific symptoms [111]. Treatment with a non-oncolytic virus is a method to take advantage of inducing a persistent immune surveillance mechanism. First, LCMV does not kill the host cell by direct cytopathic effects; therefore, viral replication is maintained until an immune response is elicited within the tumor. Induction of anti-tumor immunity by arenavirus infection depends on IFN-producing Ly6Ct monocyte cells and increased tumor-specific CD8^+^ T cells in murine oropharynx carcinoma cell-tumor-bearing mice [110]. Second, arenavirus replication cannot solely be limited by a strong IFN response [112]. Furthermore, arenaviruses do not normally induce rapid neutralizing antibodies [113]. As intratumoral replication of arenaviruses is regulated only by infiltrating CD8^+^ T cells, its replication persists for days to weeks unless CD8^+^ T cells invade the tumor. Infected cells are also eliminated by natural immunity by NK cells. Thus, persistent infection with non-oncolytic viruses is expected as an antitumor immunotherapy with limited development of neutralizing antibodies.

## 7. OV-Induced Cell Death

In addition to apoptosis, OV infection induces cell death associated with autophagy, necroptosis, and pyroptosis [114]. Understanding the cell death patterns of tumor cells due to virus infection will lead to the development of therapeutic approaches to sensitize tumor cells to OV therapy.

In apoptosis, there are death receptor and mitochondrial pathways. The Bcl-2 family is a major player in apoptosis. Inhibition of Bcl-xL releases Bad and Bax, enabling apoptosis, whereas Bcl-2 phosphorylation releases Beclin1 acting on autophagy formation. Conditionally replicating adenovirus SG235 expresses exogenous tumor necrosis factor-related apoptosis-inducing ligand (SG-TRAIL) and is able to induce apoptosis of leukemic cells via the activation of extrinsic and intrinsic apoptotic pathways [115]. VSV has a direct tumor cell lytic ability by inducing endogenous and exogenous apoptotic signals in cancer cells, but it is sensitive to IFN-γ induction and cellular antiviral response [116,117]. Many primary cancers, including chronic lymphocytic leukemia (CLL), are resistant to VSV-induced oncolysis due to overexpression of the Bcl-2 family. Therefore, blocking this Bcl-2 pathway may improve the efficacy of OV therapy. A synthetic derivative of prodiginine, Obatoclax (GX15-070), acts as a pan-Bcl-2 inhibitor. ABT-737 and its orally active analogue ABT-263 (navitoclax) are BAD-like analogues. In phase I and II trials, the therapeutic effects of Obatoclax and ABT-263 were investigated in many solid cancers and hematological malignancies, including CLL, and non-Hodgkin’s lymphoma. The combined use of VSV and ABT-737 induced autophagy and apoptotic cell death in CLL cells by inhibiting the inhibitory binding between Bcl-2 and Beclin-1 [118]. Oncolytic HSV-1 alone did not provide sufficient antitumor effects in a mouse breast cancer model, but treatment with HSV-1 ICP0 lacking KM100 in combination with an efficient ICD inducer, mitoxantrone, significantly improved the survival rate of Her2/neu TUBO-derived tumor-bearing Balb/c mice [119]. These effects were mediated by the infiltration of neutrophils and tumor antigen-specific CD8+ T cells. Depletion studies confirmed that CD8^+^ T cells, CD4^+^ T cells and Ly6G-expressing cells are essential for improved efficacy of the combination therapy [119].

Autophagy acts on both cell survival and cell death. The ability of OV to induce autophagy in cancer cells plays a role in the antitumor effects of OV [120,121,122]. In starved cells, a nutrient-sensitive mechanism that regulates the balance of AMPK and mTOR functions controls autophagy. The AMPK/mTOR pathway is not only a key during starvation, but is also a common target for several viruses that alter autophagy in host cells. In addition to mTOR, the c-Jun N-terminal kinase (JNK) is an autophagy regulator. Activated JNK phosphorylates Bcl-2 and inhibits the formation of Bcl-2/Beclin1 autophagy suppressor complexes [123]. Upon infection with HSV-based OV, Baco-1, treatment with rapamycin, which induces autophagy with an inhibitor of mTOR signaling, does not affect virus replication in permissive cells of oncolytic HSV-1. However, rapamycin resulted in a marked increase in viral replication and spread in Baco-1-resistant human esophageal carcinoma cells [124]. After infection of human oral SCC cells with γ34.5-deficient oncolytic HSV-1 RH2, autophagy was induced accompanying the aggregation of microtubule-associated protein light chain 3 (LC3) in the cytoplasm, the conversion of LC3-I to LC3-II, and the formation of autophagosomes. The autophagy inhibitors, 3-methyladenine (3MA) and bafilomycin A1, did not affect viral replication, but significantly inhibited the cytotoxicity of OV. This indicates the induction of autophagic cell death of oral SCC cells by oncolytic RH2 infection. Conversely, rapamycin increased the cytotoxicity of the OV therapy [125]. SG511-BECN is a new Ad5/11 fiber chimeric conditionally replicating adenovirus armed with the autophagy gene Beclin1, a gene that induces autophagic cell death [126]. When SG511-BECN was combined with doxorubicin, the viral infection efficiency was improved synergistically in the multidrug-resistant chronic myelogenous leukemia (CML) cell line k562/a02. Ad-cycE is a E1b-deleted oncolytic adenovirus, in which E1a gene is driven by the cyclin E promoter. The combined use of rapamycin and Ad-cycE resulted in stronger anti-tumor effects than their use alone, but the combination did not promote virus replication in normal lung cells [127]. Reovirus induced autophagy, which is required for its replication, in several cell lines, causing Atg5-Atg12 complexes, LC3 lipidation, p62 degradation, expression of acidic vesicular organelles and LC3 puncta. By knocking out autophagy-related genes, the expression of Atg3 and Atg5, but not Atg13, was found to be required for reovirus replication. Preclinical studies performed on oncolytic paramyxoviruses, including NDV, measles virus, Sendai virus and Morbillivirus, have demonstrated that the PI3K/Akt/mTOR/p70S6K pathway plays a critical role in autophagy and ICD in cancer models [122].

## 8. Improvement of Virus Replication

Among the actions of OV, alteration of tumor immunity is often emphasized, but viral replication in tumors is essential for successful outcomes. In general, cancer cells have defective IFN signaling pathways and are susceptible to oncolysis by the IFN-inducing viruses, but some tumors resist VSV-induced oncolysis. Therefore, it is necessary to develop methods to promote viral transcription or to reduce the production of IFN and IFN-inducible factors that inhibit viral replication. HNSCC cell lines, SCC25 and SCC15 expressing IFNα/β-reactive antiviral genes IRF9, TRF-7 and MxA1 constitutively, are resistant to VSV compared with VSV-sensitive SW579 cells. When VSV-resistant cells were treated with VSV and JAK1/2 inhibitors (JAK inhibitor I and ruxolitinib), the mRNA level of these genes decreased and the sensitivity of cells to VSV cytotoxicity was restored [128]. JAK inhibitor I and ruxolitinib are promising concomitant drugs that can eliminate the VSV resistance of HNSCC. Drug-induced destabilization of microtubules inhibits IFN mRNA translation and IFN expression and secretion, and also induces bystander killing through the production of virus-induced inflammatory and cytotoxic cytokines (TNF-α, IL-6, IL-8, IP-10, and IL-29), accompanying a significant increase in the proportion of polynuclear cells [129]. Colchicine destabilizes microtubules, whereas paclitaxel stabilizes microtubules. Microtubule-destabilizers, colchicine and vinorelbine, can sensitize cancer cells to VSV and improve their therapeutic effects on VSV-resistant cancer cells. Paclitaxel combined with IFN-sensitive Maraba virus also increased virus production in breast cancer cells and increased the antitumor effects of the virus [130].

Histone deacetylases (HDACs) significantly influence virtually all cellular processes by affecting the activities of histones and numerous proteins, including transcription factors, chaperones, and regulators of DNA repair, replication and transcription. HDAC deregulation has been implicated in the promotion of both carcinogenesis and tumor progression [131], thereby promoting the development of a number of histone deacetylase inhibitors (HDACIs) with wide-ranging anti-cancer properties [132]. HDACI-induced cell death is often immunogenic and increases antitumor immunity. HDACIs currently approved by the FDA are vorinostat, romidpsin, and belinostat for the treatment of T-cell lymphoma [133]. HDACIs suppress the antiviral activity of tumor cells and improves OV replication and spread [133,134]. Pretreatment with valproic acid (VPA) suppressed the transcription of IFN-stimulated anti-viral genes, such as signal transducers and activators of transcription 1 (STAT1), protein kinase R (PKR) and promyelocytic leukemia (PML), thereby increasing HSV gene expression, propagation and cytotoxicity [135]. VPA inhibited NK cell-mediated cytotoxicity both by down-regulating granzyme B and perforin and by abrogating NK cell-dependent production of IFN-γ through the inhibition of the STAT5/T-BET pathway [136]. The HDACI trichostatin A (TSA) improved the replication and cytotoxicity of ICP34.5-deficient HSV-1 variant R849 in oral SCC cells. By increasing the acetylation level of NF-κB p65 subunit, TSA promoted the nuclear translocation and activity of NF-κB. It increased the expression of p21, leading to G1 cell cycle arrest and antitumor effects [137]. Similarly, in the treatment of glioblastoma (GBM) and colorectal cancer using the multi-attenuated HSV-1 mutant G47D, TSA improved the antitumor effects synergistically, with cyclin D1 blockade and VEGF inhibition [138]. A panel of HDACIs was tested in vitro for their ability to increase the replication of ICP34.5-deleted oncolytic HSV-1 in breast cancer-derived cell lines. Pan-HDACIs or HDACIs targeting class I HDACs were more effective than those for class II HDACs or those selective for a particular HDAC. A number of HDACIs, including romidepsin (also named FK228 or depsipeptide), TSA, sodium butyrate and VPA, improved the infectivity and transduction ability of adenovirus-based gene transfer vectors by increasing the expression of the cellular coxsackievirus and adenovirus receptor (CAR) and integrin alpha V in various solid and hematological cancer-derived cell lines. Furthermore, HDACIs can increase the anti-cancer efficacy by increasing the transfer and transcription of the TRAIL gene, activating the TRAIL-mediated pathway and reducing expression of the anti-apoptotic proteins Bcl-xL and c-FLIP [139,140]. Among the HDACIs, TSA was the most effective at increasing vaccinia virus replication, spread and killing activity, mainly though inhibition of the IFN anti-viral response [141]. HDACIs increased VSVd51 replication and activation of the intrinsic apoptotic pathway by inhibiting the expression of IFN and IFN- and IFN-inducible genes such as IRF3, IRF7, and MXI [141]. The promotion of VSV oncolysis by vorinostat in prostate cancer cells was traced back to the reversible induction of NF-κB signaling through increased acetylation, nuclear translocation and DNA binding activity of the NF-κB subunit RELA/p65 [142].

Hexamethylene bisacetamide (HMBA) is a low-molecular-weight polar compound with potent anti-cancer and cell differentiation activities [143,144]. HMBA induces terminal differentiation via upregulation of HMBA inducible protein 1 (HEXIM1) [145]. Although clinical studies were conducted as a promising candidate for differentiation induction therapy of leukemias, dose- dependent toxicity of thrombocytopenia due to a short biological half-life was noted and further clinical trials were discontinued [143,146]. The virus protein VP16 of HSV-1 activates the transcription of the immediate early (IE) genes through specific sequence elements in the IE promoters [147]. Complete deletion of the VP16 activation domain resulted in significant disruption of the normal global patterns of regulated HSV gene expression. This disruption was largely overcome in the presence of HMBA. Indeed, in the presence of HMBA, a dose-dependent increase in HSV-1 yield was observed in HEp-2 epidermal cells and IMR-32 neuronal cells [148]. HMBA was also a potent inducer of HIV production in chronically infected cells. HMBA transiently activates the PI3/K pathway, which leads to the phosphorylation of HEXIM1 and the subsequent release of active positive transcription elongation factor b (P-TEFb) from its transcriptionally inactive complex with HEXIM1 and 7SK small nuclear RNA (snRNA). As a result, P-TEFb is recruited to the HIV promoter to stimulate transcription elongation and viral production [149]. By this mechanism, transcription of OV may be increased to improve virus replication. Indeed, in human oral SCC cells infected with γ34.5 gene-deficient HSV-1 R849, HMBA increased the mRNA expression of ICP0, ICP4 and ICP27 genes, and increased viral replication and viral cytotoxicity; however, it did not affect virus replication in normal epithelial cells. In animal studies, HMBA did not cause HSV-1 infection in organs, such as the lung, liver and spleen, but it increased expression of viral antigens in the tumors, suppressed tumor growth and prolonged animal survival [150]. As the anticancer effects and adverse events in clinical usage of HMBA were clarified, it should be considered as a sensitizing drug for OV therapy.

## 9. Delivery and Spread of OV in Tumors

In solid tumors, there are a series of barriers that OVs need to overcome to reach the tumor cells. Viral spread in the TME is hampered by the interstitial liquid pressure provided by the lack of functional lymphatic vessels and dense extracellular matrix, which can impair viral infiltration [151]. Even with direct intratumoral injection, many administered viruses are immediately lost from the site of administration due to uncertainties of injection and because the virus is less likely to penetrate the surrounding tissue. In addition, OVs can induce strong innate responses, together with wide-spread antiviral immunity, preexisting circulating antibodies and blood factors such as the coagulation factors [23]. In organ culture of oral mucosa, the primary infection site, replication of HSV-1 was restricted to only bioactive cells, suggesting tumor tissues are not as permissive as cells in culture [152]. In recurrent lesions of HNSCC, scar tissue due to surgery and radiation may make viral infiltration difficult or cause virus to leak from the necrotic area. 

A novel carrier system to deliver oncolytic adenovirus by tumor cell-derived microparticles has been developed. This prevents the antiviral effects of circulating antibodies and induces highly efficient cytolysis of tumor cells [153]. MDSCs are tumor-tropic, mobilized and capable of invading tumor lesions, and therefore can be used as a carrier in delivering VSV [154]. Similarly, cytokine-induced cytotoxic cells, neural stem cells, mesenchymal stem cells and irradiated tumor cells have been used for virus delivery [155,156,157]. Nanoparticles, liposomes, polyethylene glycol (PEG) and polymeric particles have also been employed to transport OVs from the systemic circulation to cancer cells [158,159,160]. In particular, synthetic nanoparticle-coated OVs survive for long periods of time and are resistant to viral clearance by antibodies. Other promising strategies are ultrasound (US) and magnetic drug targeting systems [161,162,163].

Cell fusion can be used to spread the virus to neighboring cells to overcome the restricted diffusion of the progeny virus within the TME [164,165,166] There are natural fusogenic viruses, but fusogenic viruses have also been created. Examples of natural viruses include NDV, Sendai virus, respiratory syncytial virus (RSV) and measles [166]. In HSV-GALV, oncolytic HSV is used as the backbone and a cell-fusible gibbon ape leukemia virus (GALV) fusion protein was introduced to improve the oncolytic potential [167,168].

Ultrasonic energy has been used for diagnostic imaging and physical therapy [169]. High US intensities can produce thermal effects enabling the adsorption of heat due to acoustic energy by tissue, which is employed by high-intensity focused US surgery or US-based physiotherapy. Alternatively, low-intensity US is not harmful and can elicit biological effects, which are implicated in the formation of temporary pores (named sonoporation) in the plasma membrane and vasculature, leading to the diffusion or extravasation of the drug or gene at the sonoporation site [170,171,172]. The efficiency of sonoporation can be markedly improved by the presence of microbubbles, which are stabilized gas microbubbles originally developed as US contrast agents for medical imaging. In response to US exposure, these microbubbles can expand and collapse creating cavitation evens and consequently micropumping, microstreaming and even shock waves, leading to formation of small pores. When oral SCC cells in culture were inoculated with oncolytic HSV-1 RH2 and then exposed to US in the presence of microbubbles, the number of plaques caused by entered virions markedly increased, suggesting that sonoporation bypassed the adsorption process of the virus to establish infection [173,174]. This pore formation to allow entry of virus particles lasted approximately 3 min after US irradiation [175]. In oral SCC xenografts in mice, tumor growth was markedly suppressed by HSV-1 RH2 in combination with US, especially with microbubbles [174]. HNSCC has the advantage in that it is easy to irradiate by US from the body surface. By the simultaneous use of the sensitizing microbubbles, sonoporation is a promising method to efficiently infect tumor cells with OV at the site of virus injection.

## 10. Conclusions

Surgery, radiotherapy and chemotherapy have been established as effective means for HNSCC. However, following the introduction of ICIs, the effectiveness of immunotherapy is expected, especially in advanced disease. The purpose of treatment with ICs that blocks an inhibitory molecule of tumor immunity is to restore the anti-tumor immunity against TSAs/TAAs originally possessed by the patient. However, not all patients possess strong antitumor immune activity and it is necessary to induce immunostimulatory inflammation to improve the immunosuppressed state in the TME. Tumor OV infection can recruit inflammatory cells to activate innate immunity, recognize TSAs/TAAs and activate passive immunity. Although the development of CAR-T therapy is advancing and there are attempts to target multiple tumor antigens, it is difficult to identify individually mutated TSAs to be targeted by CAR-T cells. However, OV therapy has the ability to stimulate immunity to such diverse tumor antigens. The modified OV incorporates immunostimulatory genes, but unless the virus is able to replicate in tumor cells, expression of these foreign genes is not expected. In order to achieve successful OV therapy, it is important to improve the efficiency of virus delivery and to enable the transcription of viral genes in tumor cells.

## Figures and Tables

**Table 1 ijms-21-07073-t001:** Clinical trials of oncolytic virotherapy for HNSCC.

Virus Type	Virus Name	Clinical Phase	Number of Patients	Route of Administration	Co-Therapy	Type of Cancer	Ref
HSV-1 (JS1 strain)	T-Vec	I/II	17	i.t.	RT+cisplatin	HNSCC, stage III/IV	[30]
		Ib	36	i.t.	pembrolizumab	HNSCC, recurrent, metastatic	[31]
HSV-1 (HF strain)	HF10	I	17	i.t.	—	breast cancer, HNSCC, pancreatic cancer, recurrent and non-resectable	[32]
Vaccinia virus (Lister strain)	GL-ONCI	I	19	i.v.	RT+cisplatin	HNSCC, locoregionally advanced	[33]
	MVA-EL	I	16	intradermal	—	nasopharyngeal carcinoma, EB positive	[34]
Adenovirus type 5	ONYX-015	II	37	i.t.	—	HNSCC, recurrent	[35]
	AdGV.EGR.TNF.11D	I	14	i.t.	RT+5FU +hydroxyurea	HNSCC, irradiated, unresectable, recurrent	[36]
Reovirus type 3 (Dearing strain)	REOLYSIN	I/II	31	i.v.	carboplatin, paclitaxel	solid tumors including HNSCC, heavily pretreated	[37]

RT, radiotherapy; i.t., intratumoral; i.v., intravenous.

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
