# Peer review of "Efficient Delivery and Replication of Oncolytic Virus for Successful Treatment of Head and Neck Cancer"

_ijms, 2020, doi:10.3390/ijms21197073_

Round 1
Reviewer 1 Report
In this review, Hamada and Yura describe the available results about the application of oncolitic virus therapy for the treatment of head and neck cancer. In gerneral it is an interesting paper focused on a state of the art anty cancer therapy of great interest for the Readers of IJMS.
I think the manuscript would benefit of few minor corrections and widenings:
1- Authors sould clearly ad briefly describe the experimental settings of the studies here reported, since they are sometimes missing. For example, in case of an in vitro study, authors should explain the cell model used (whether animal, human, primary or immortalized lines). In case of in vivo studies or clinical trials, authors should describe the sample size, the gender of participants and the parameters used to evaluate the patient status.
2- Is the tumor-related downregulation of INF pathway the only factor driving the virus tropism for cancer cells? Intuitively, it doesn't seem very strong as a rational. Aothors should describe, whenever existing, the others factors directing the virus activity and, alternatively, driving the choice towards an intra-tumor injection.
Author Response
September 18, 2020
International Journal of Molecular Sciences
Assistant Editor
Dear Prof. Dr. Candice Yang
We have studied the reviewer’s comments. Most of them are reasonable and helpful. We are returning our manuscript entitled “Efficient delivery and replication of oncolytic virus for successful treatment of head and neck cancer” after revisions made in compliance with referee’s comments.
Reviewer1
In this review, Hamada and Yura describe the available results about the application of oncolitic virus therapy for the treatment of head and neck cancer. In gerneral it is an interesting paper focused on a state of the art anty cancer therapy of great interest for the Readers of IJMS.
I think the manuscript would benefit of few minor corrections and widenings:
1- Authors sould clearly ad briefly describe the experimental settings of the studies here reported, since they are sometimes missing. For example, in case of an in vitro study, authors should explain the cell model used (whether animal, human, primary or immortalized lines). In case of in vivo studies or clinical trials, authors should describe the sample size, the gender of participants and the parameters used to evaluate the patient status.
Response: Thank you for your positive comments to our paper. Following the suggestion, we have added description for the experimental settings and a table summarizing the clinical trials in the field of head and neck cancer in which virus name, clinical phase, number of patients, stage of the disease have been described.
2- Is the tumor-related downregulation of INF pathway the only factor driving the virus tropism for cancer cells? Intuitively, it doesn't seem very strong as a rational. Aothors should describe, whenever existing, the others factors directing the virus activity and, alternatively, driving the choice towards an intra-tumor injection.
Response: Following the suggestion of the reviewer, we have added the sentences explaining other pathways to infect tumor cells selectively and an example of the administration route of HSV-1 (lines 77-82).
Yours sincerely,
Masakazu Hamada DDS, PhD
Department of Oral and Maxillofacial Surgery, Osaka University Graduate School of Dentistry
Reviewer 2 Report
The present manuscript by Hamada et al. entitled “Efficient delivery and replication of oncolytic virus for successful treatment of head and neck cancer” describes the mechanisms of action of the oncolytic virus therapy and summarizes clinical studies that used this therapy as a treatment for head and neck cancer.
Minor improvements needed:
Many abbreviations need to be described.
Line 68-200. I think the data could be better organized mentioning first the studies performed using cell lines, then animal models, and finally the studies in humans. I would suggest a table reviewing the information in section 2 (Current OV Therapy for HNSCC). For each virus, the columns could be the family, genus, type of genome, clinical studies, number of patients, type of cancer, whether it was the OV therapy alone or in combination, etc. everything that could help to understand the article. The most common description between the oncolytic viruses the better, even if there is not a table.
In my opinion, a figure would be helpful
Given the expertise of the author in the field, I would suggest including a broader discussion of the disadvantages of this therapy, the side effects and the limitations.
Author Response
September 18, 2020
International Journal of Molecular Sciences
Assistant Editor
Dear Prof. Dr. Candice Yang
We have studied the reviewer’s comments. Most of them are reasonable and helpful. We are returning our manuscript entitled “Efficient delivery and replication of oncolytic virus for successful treatment of head and neck cancer” after revisions made in compliance with referee’s comments.
Reviewer2
The present manuscript by Hamada et al. entitled “Efficient delivery and replication of oncolytic virus for successful treatment of head and neck cancer” describes the mechanisms of action of the oncolytic virus therapy and summarizes clinical studies that used this therapy as a treatment for head and neck cancer.
Minor improvements needed:
Many abbreviations need to be described.
Response: Following the suggestion, we have checked and spell out abbreviations.
Line 68-200. I think the data could be better organized mentioning first the studies performed using cell lines, then animal models, and finally the studies in humans. I would suggest a table reviewing the information in section 2 (Current OV Therapy for HNSCC). For each virus, the columns could be the family, genus, type of genome, clinical studies, number of patients, type of cancer, whether it was the OV therapy alone or in combination, etc. everything that could help to understand the article. The most common description between the oncolytic viruses the better, even if there is not a table.
Response: Following the suggestion of the reviewer, we have added the table 1 including virus type, clinical phase, number of patients, and type of cancer,
In my opinion, a figure would be helpful
Response: Figures may help to better understand mechanisms such as the structure of oncolytic viruses, signaling systems associated with virus infection, and anti-tumor immunoactivation pathways, but this review does not focus on such specific mechanisms, so figures are not included.
Given the expertise of the author in the field, I would suggest including a broader discussion of the disadvantages of this therapy, the side effects and the limitations.
Response: Following the suggestion of the reviewer, we have added a new section with the title “Safety and limitation of OV therapy” (lines 213-231).
Yours sincerely,
Masakazu Hamada DDS, PhD
Department of Oral and Maxillofacial Surgery, Osaka University Graduate School of Dentistry